# Ten Questions on Using Lung Ultrasonography to Diagnose and Manage Pneumonia in Hospital-at-Home Model: Part II—Confounders and Mimickers

**DOI:** 10.3390/diagnostics15101200

**Published:** 2025-05-09

**Authors:** Nin-Chieh Hsu, Yu-Feng Lin, Hung-Bin Tsai, Charles Liao, Chia-Hao Hsu

**Affiliations:** 1Department of Internal Medicine, National Taiwan University Hospital, Taipei 100, Taiwan; chesthsu@gmail.com (N.-C.H.);; 2Division of Hospital Medicine, Department of Internal Medicine, Taipei City Hospital Zhongxing Branch, Taipei 103212, Taiwan; 3Taiwan Association of Hospital Medicine, Taipei 100225, Taiwan; 4Department of Internal Medicine, Stanford University School of Medicine, Stanford, CA 94305, USA; 5Department of Orthopedics, Kaohsiung Medical University Hospital, Kaohsiung 80756, Taiwan; 6College of Medicine, Kaohsiung Medical University, Kaohsiung 80708, Taiwan

**Keywords:** pneumonia, point-of-care, ultrasonography, lung ultrasonography, hospital-at-home, diagnosis, treatment, consolidation, air bronchogram, interstitial lung disease, heart failure

## Abstract

The hospital-at-home (HaH) model offers hospital-level care within patients’ homes and has proven effective for managing conditions such as pneumonia. The point-of-care ultrasonography (PoCUS) is a key diagnostic tool in this model, especially when traditional imaging modalities are unavailable. This review explores how PoCUS can be optimized to manage pneumonia in HaH settings, focusing on its diagnostic accuracy in patients with comorbidities, differentiation from mimickers, and role in assessing disease severity. Pulmonary comorbidities, such as heart failure and interstitial lung disease (ILD), can complicate lung ultrasound (LUS) interpretation. In heart failure, combining lung, cardiac, and venous assessments (e.g., IVC collapsibility, VExUS score) improves diagnostic clarity. In ILD, distinguishing chronic changes from acute infections requires attention to B-line patterns and pleural abnormalities. PoCUS must differentiate pneumonia from conditions such as atelectasis, lung contusion, cryptogenic organizing pneumonia, eosinophilic pneumonia, and neoplastic lesions—many of which present with similar sonographic features. Serial LUS scoring provides useful information on pneumonia severity and disease progression. Studies, particularly during the COVID-19 pandemic, show correlations between worsening LUS scores and poor outcomes, including increased ventilator dependency and mortality. Furthermore, LUS scores correlate with inflammatory markers and gas exchange metrics, supporting their prognostic value. In conclusion, PoCUS in HaH care requires clinicians to integrate multi-organ ultrasound findings, clinical context, and serial monitoring to enhance diagnostic accuracy and patient outcomes. Mastery of LUS interpretation in complex scenarios is crucial to delivering personalized, high-quality care in the home setting.

## 1. Introduction

The hospital-at-home (HaH) model delivers acute care in patients’ homes, offering hospital-level diagnostics, monitoring, and treatments, including some advanced therapies [1,2,3]. It is effective for conditions like pneumonia, heart failure, COPD, urinary tract infections, and skin infections [3,4,5].

Delivering HaH care requires point-of-care services like portable diagnostics, remote monitoring, medication therapies, and telemedicine [5,6,7]. Among diagnostic tools, point-of-care ultrasonography (PoCUS) is key [7]. In hospitals, PoCUS complements imaging; in HaH, it often serves as a primary or sole diagnostic method [8,9,10,11].

The literature supports the use of PoCUS in managing COVID-19 pneumonia, informing similar home-based care approaches [12,13]. In Part I of our review, we summarised current PoCUS practices for diagnosing pneumonia in the ward and emergency department settings (Table 1). We reviewed key techniques and primary sonographic patterns that may be adapted for home care settings and recommended a bilateral anterior, lateral, and posterior zone-based examination protocol [14]. This review further examines key questions to optimize the role of PoCUS in managing pneumonia within the HaH model.

## 2. Question 4: Do Pulmonary Comorbidities Affect the Accuracy of Ultrasound Diagnosis for Pneumonia?

Concerns about pulmonary comorbidities affecting ultrasound diagnosis of pneumonia are two-fold. First, lung ultrasound relies partly on artifacts rather than direct visualization of tissue changes. Second, its high sensitivity—compared to chest X-ray—in detecting interstitial and alveolar changes can amplify “noise”, complicating severity assessment.

Solutions are also two-fold. First, additional information beyond lung and thoracic ultrasound is essential, including left ventricular dynamics, right ventricular size, jugular vein distension, inferior vena cava size, and their variations. Second, efforts should focus on identifying ultrasound patterns more specific to pneumonia versus other pathologies.

Here, we discuss two common scenarios in the HaH setting, review current evidence, and propose some practical solutions.

### 2.1. Pneumonia Diagnosis in Patients with Heart Failure

Complex patients often present with multifactorial conditions that defy simplified diagnostic approaches. In the landmark study validating the lung ultrasound (LUS) protocol, patients with mixed diagnoses were excluded from the final analyses [15]. As a result, those with concomitant cardiogenic pulmonary edema and pneumonia may not receive a comprehensive diagnosis using traditional protocols. While the BLUE protocol represents a significant advancement in bedside ultrasound, its limitation lies in relying solely on sonographic findings without integrating clinical context. In real-world practice, clinicians are rarely, if ever, blind to patients’ subjective symptoms, physical examination findings, and medical history, all of which are essential for accurate diagnosis and management.

In patients with heart failure or suspected heart failure, several PoCUS findings are readily accessible and demonstrate high sensitivity and specificity, even when used by non-experts [16,17]. Focused cardiac ultrasound (FoCUS) has been endorsed by the American Society of Echocardiography [18], and resources aimed at training non-cardiologists in FoCUS are becoming increasingly widespread [19,20,21]. In a large cohort of patients with Chagas disease, FoCUS demonstrated satisfactory validity and reliability in assessing left and right ventricular size and function, even when performed by an observer blinded to clinical and standard echocardiographic data [22].

Patients with well-controlled heart failure typically exhibit minimal interstitial lung edema and are less likely to present with multiple B-lines on LUS [23,24]. In contrast, patients with acute decompensated heart failure (ADHF) commonly display bilateral or diffuse B-lines [25]. In such cases, additional assessment of intravascular volume status or organ congestion is often necessary. Studies in ADHF patients have shown that combining LUS findings with IVC collapsibility offers valuable prognostic information—both the number of B-lines and IVC diameter and collapsibility were independently predictive of outcomes [24]. Therefore, in patients with undifferentiated multifocal B-lines, PoCUS evaluation of IVC dynamics can aid clinical decision-making.

Venous congestion associated with acute kidney injury can be identified using PoCUS [26], and this evaluation method has gained popularity through the VExUS protocol [27]. Organ congestion, particularly in the context of right heart dysfunction, is a hallmark of heart failure. A similar Doppler-based protocol has been validated for estimating right atrial pressure [28]. In patients with acute heart failure, the VExUS score derived from Doppler waveforms of the hepatic, portal, and renal veins has become well-known for assessing venous congestion [29]. Notably, the VExUS score has demonstrated superior predictive value for in-hospital mortality compared to other markers of venous congestion, such as right atrial function and IVC diameter. A recent study showed that VExUS scores of 0 or 1 effectively ruled out elevated right atrial pressure and pulmonary capillary wedge pressure [30]. Therefore, in patients with undifferentiated multifocal B-lines, assessment of VExUS grade, with or without renal vein Doppler, can provide valuable diagnostic information.

We proposed a flow diagram for evaluating pneumonia in patients with a history of heart failure (Figure 1). This stepwise approach incorporates lung ultrasound, FoCUS, IVC assessment, and VExUS grading. Notably, in cases of overt right ventricular dilation on FoCUS, IVC diameter and collapsibility may be unreliable due to elevated right-sided heart pressures. In such scenarios, the VExUS grade can provide valuable diagnostic clues. It is also important to note that pneumonia and pulmonary edema can coexist; thus, favoring one does not preclude the presence of the other.

Current evidence on diagnosing pneumonia in patients with heart failure or those at risk of pulmonary edema remains limited. Ultrasound has proven to be a valuable tool for predicting and monitoring the course of COVID-19 pneumonia. The impact of cardiac dysfunction on ultrasound findings in COVID-19 patients has been investigated, and notably, the presence of cardiac dysfunction did not significantly alter LUS results [31]. This suggests that in the absence of congestion or fluid overload, heart failure minimally confounds LUS interpretation. Another study evaluated patients with acute myocardial infarction who are at risk for cardiogenic pulmonary edema. Using computed tomography (CT) as the gold standard, LUS demonstrated a sensitivity of 92% and specificity of 96% for diagnosing pneumonia, outperforming chest X-ray [32].

### 2.2. Pneumonia Diagnosis in Patients with Interstitial Lung Disease

LUS shows promise in detecting ILD in at-risk populations, such as those with rheumatoid arthritis [33], Sjögren’s syndrome [34], and systemic sclerosis [35,36]. Due to its high sensitivity, LUS has a high negative predictive value for ILD in these groups [37,38]. Moreover, LUS can be used to assess the extent and severity of ILD [39]. In a cohort of individuals with rheumatoid arthritis, the positive criteria for ILD using a 14-zone LUS protocol included the presence of at least 10 B-lines or bilateral thickened and fragmented pleura [37]. A meta-analysis of eight studies comprising 868 participants reported that LUS detection of B-lines yielded a sensitivity of 93% and specificity of 61% for ILD diagnosis [35]. Another useful LUS feature of ILD is pleural irregularity (PI) [40]. In patients with Sjögren’s syndrome, a PI postero-inferior score cut-off of 15 provided a sensitivity of 86.6% and specificity of 84.2% for ILD diagnosis [40].

Patients with ILD may experience acute worsening of dyspnea due to acute exacerbation [41], which can be triggered by various potential factors [42]. Although pulmonary viral or bacterial infections should be considered during exacerbations, evidence suggests they are not common causes. A single-center cohort study conducted before and after the COVID-19 pandemic reported that viral infections were rare, with only two cases of SARS-CoV-2, and bacterial infections were infrequently detected [43]. Another study in Japan analyzed nasopharyngeal specimens from patients with acute exacerbation of interstitial lung disease using the FilmArray assay and suggested that viral infections were not a frequent cause of these exacerbations [44].

LUS use in patients with ILD presents a challenging scenario in the HaH setting. Since pneumonia is a common community-acquired infection [45], it is not uncommon among patients with ILD. Differentiating pre-existing B-lines associated with ILD from new B-lines due to pneumonia can be difficult. However, certain clues may aid in this distinction. One study comparing LUS with high-resolution CT found that all patients exhibited diffuse bilateral B-lines [46], suggesting that unilateral or significantly focal B-line patterns should raise suspicion for pneumonia. The distance between adjacent B-lines was also found to correlate with CT patterns: diffuse B-lines with a narrow distance (approximately 3 mm) correlated with ground glass opacity (GGO), while those with a wider distance (approximately 7 mm) correlated with extensive fibrosis and honeycombing [46]. These B-line patterns, termed B3 and B7 lines, were further validated in another cohort, demonstrating significant associations with alveolar and interstitial patterns, respectively [47].

In patients without a known history of ILD, the detection of bilateral pleural thickening or irregularity on LUS raises suspicion for ILD [37,38,39]. A high-frequency linear probe can further aid in delineating pleural abnormalities (Figure 2). In patients with known ILD, efforts should be made to identify whether a specific LUS pattern of pneumonia exists. A 12-zone, 14-zone, or even comprehensive intercostal space scanning may be required to detect consolidation. Any consolidation on LUS may support the diagnosis of superimposed pneumonia. In the absence of consolidation on LUS, predominant B3 lines favor an alveolar process corresponding to GGO on chest CT, possibly caused by pneumonia. In contrast, when most B-lines are B7, pneumonia is less likely, although exclusion remains difficult (Figure 3).

The new paradigm of PoCUS encompasses problem-oriented multi-organ evaluation. In a landmark study for patients with acute respiratory symptoms, a focused heart, lung, and deep vein ultrasound evaluation identified missed life-threatening conditions [48]. A subsequent randomized controlled study reported that 88% of patients who underwent the multi-organ PoCUS evaluation, compared to 63.7% in the standard care group, obtained a correct diagnosis within four hours of arriving at the emergency room [49]. Home care physicians should remember that ultrasound, in their hands, is more than an organ-specific diagnostic tool. It allows you to gather comprehensive information to guide precise treatment for complex patients.

### 2.3. Pneumonia Diagnosis in the Presence of Pleural Effusion

It is important to note that both A-line and B-line artifacts rely on a pleural surface that is parallel and close to the chest wall and the ultrasound probe. In the presence of pleural effusion, the visceral pleura is displaced away from the probe, becoming non-parallel to the chest wall and non-perpendicular to the ultrasound beam, which prevents the formation of A-lines. Furthermore, subpleural interlobular septa are no longer aligned with the ultrasound beam, hindering the generation of B-lines. Several other horizontal or vertical artifacts may appear but are not associated with pleural or alveolar pathology. Therefore, in cases of pleural effusion, only subpleural consolidations and air bronchograms remain reliable sonographic signs for diagnosing pneumonia.

## 3. Question 5: Do Other Differential Diagnoses Mimic the Ultrasound Patterns of Pneumonia?

Previous studies have shown that 5–17% of patients with initial suspicion of infectious pneumonia have other pulmonary diseases [50,51]. Clues for alternative diagnoses at the time of initial presentation include bilateral consolidations, migratory infiltrates, and a discrepancy between extensive abnormalities on image findings and minor clinical symptoms. Over the course of the disease, therapy-resistant “pneumonia” may raise suspicion for an alternative diagnosis. A number of non-infectious conditions, including atelectasis, lung contusion, neoplastic lesions, pulmonary edema, pulmonary embolism, drug-induced pneumonitis, diffuse alveolar hemorrhage syndromes, cryptogenic organizing pneumonia, and acute eosinophilic pneumonia, may present in a similar way and mimic community-acquired pneumonia [52].

### 3.1. Atelectasis

Atelectasis typically appears as consolidation on lung ultrasound (LUS) due to the loss of aeration within affected segments or lobes. This finding often resembles the air bronchogram pattern seen in pneumonia. Although atelectasis is common in patients with impaired airway clearance, it is less frequently reported in clinical protocols. For instance, atelectasis is not listed as a final diagnosis in the flowchart of the BLUE protocol. One possible explanation is that atelectasis may be transient and reversible once the airway is cleared and patency is restored. However, in a recent study in the ICU, 69 of 120 patients had a diagnosis of atelectasis by LUS [53].

Some characteristics are helpful in recognizing atelectasis. Air bronchograms result from trapped air within the bronchial tree, appearing as hyperechoic circles on ultrasound. In atelectasis, these air bronchograms are static and do not move with respiration. Static air bronchograms, as opposed to dynamic ones that move with respiration, are a hallmark feature of atelectasis (Figure 4) [53,54]. Color Doppler imaging can assist in evaluating suspected static air bronchograms. If pulsatile flow is absent, pneumonia can be ruled out, supporting a diagnosis of atelectasis, which has been reported to have high sensitivity in this context [55,56].

A lung pulse and B-lines may also be present in atelectasis. When atelectasis is due to bronchial obstruction, lung sliding is typically absent. Therefore, the combination of absent lung sliding with a preserved lung pulse is highly suggestive of atelectasis [57,58]. Another useful sign of atelectasis is the presence of consolidation as well as traction of anatomical landmarks associated with volume reduction. For example, a right middle lobe collapse drives a right deviation of the heart (Figure 5).

### 3.2. Lung Contusion

Lung ultrasound findings of lung contusion include an increased number of B-lines in the affected regions and the presence of consolidation. In patients with a history of blunt chest trauma, multiple B-lines have a sensitivity of 94.6%, while consolidation demonstrates a specificity of nearly 100% for diagnosing lung contusion [59]. Compared to conventional chest X-ray, ultrasound can detect lung contusion earlier and with significantly higher sensitivity and specificity [60].

However, the LUS patterns of lung contusion overlap with those of pneumonia, making it difficult to distinguish between the two based on ultrasound alone. Therefore, careful assessment of the chest wall for signs of trauma, thorough history-taking, and maintaining a high index of suspicion in relevant clinical contexts is essential for accurate diagnosis.

### 3.3. Cryptogenic Organizing Pneumonia or Eosinophilic Pneumonia

Organizing pneumonia is a pattern of lung tissue repair following injury, which may be cryptogenic or secondary to a specific lung insult, and is observed histopathologically in various clinical contexts [61,62]. Cryptogenic organizing pneumonia (COP), with no identifiable cause, is classified as a form of idiopathic interstitial pneumonia (IIP) and typically presents with an acute or subacute course, along with histological features of acute lung injury [63]. Parenchymal consolidation is the most common imaging finding, often accompanied by an air bronchogram, and these consolidations frequently vary in extent and location throughout the disease course, commonly affecting the peripheral and basal lung regions. Clinically, cough, fever, and dyspnea are the most frequent manifestations. Imaging may reveal single or multiple consolidations, lung nodules, migratory opacities, the reversed halo sign, and areas of ground-glass opacity. Computed tomography is usually required for diagnosis.

Reports of LUS findings in COP are limited, as its features often resemble those of pneumonia. COP should be suspected when LUS follow-up reveals migratory or shifting subpleural consolidations. Consolidations that persist or change in size and extent without corresponding to the clinical treatment response may suggest COP or other IIP.

Acute eosinophilic pneumonia (EP) is a rare, idiopathic condition characterized by the accumulation of eosinophils in the pulmonary alveoli and interstitial lung tissue. In a study involving 22 patients with acute EP who underwent lung ultrasound (LUS) examinations, all patients demonstrated multiple diffuse bilateral B-lines, along with preserved lung sliding [64]. The disease presents acutely, and some patients may rapidly progress to respiratory failure. Patients with hypoxemia and bilateral lung consolidation should be excluded from consideration for HaH care models.

### 3.4. Neoplastic Lesions

Lung cancer in contact with the pleura can present as subpleural consolidative lesions mimicking pneumonia. In patients with minimal airway symptoms, it is often an incidental finding. Unlike COP, EP, or other IIPs, which are typically multifocal and bilateral, lung cancer usually presents initially as a unilateral focal consolidation. In clinical practice, clinicians have no reliable method to differentiate between pneumonia and lung cancer. Even with contrast-enhanced LUS, a large cohort study reported a similar diagnostic challenge [65]. Elastography, which assesses tissue stiffness in lung lesions, may be a promising tool for differentiating benign from malignant lesions [66,67]. However, it is not widely available and has limited applicability in the PoCUS setting. Though LUS guidance can be helpful, central lesions located far from the pleura require an endobronchial approach [68].

Several key features suggestive of lung cancer warrant attention. Malignant signs such as rib destruction, intercostal muscle infiltration, and obliteration of extrapleural fat are rarely observed in pneumonia, except in cases complicated by abscess formation or septic emboli [69,70,71]. Assessment of mass effect [72] or external invasion in consolidative lesions can be particularly helpful in reminding the sonographer of the possibility of malignancy-associated consolidation (Figure 6).

## 4. Question 6: Do Ultrasound Findings Correlate with Pneumonia Severity?

Several studies have explored extending the use of PoCUS beyond initial diagnosis to include serial follow-up in patients with pneumonia. One study applied LUS using a standardized protocol at baseline and then every other day for the first 15 days to monitor COVID-19 pneumonia in the ICU [73]. Among those with an increasing LUS score between day 1 and day 7, 92% required invasive ventilation, compared to 57% of those with a decreasing LUS score. These findings suggest that dynamic changes in LUS, referenced against baseline values, are associated with disease severity and may reflect either progression or recovery. Another study also demonstrated that follow-up LUS is a valuable tool for monitoring disease progression and predicting the impending development of severe illness in patients with COVID-19 [74]. In intubated patients, LUS scores are closely correlated with disease trajectory. Patients who were successfully extubated showed a decrease in LUS scores, which were lower than at the time of intubation. In contrast, a patient who died from refractory hypoxemia showed a persistently increasing LUS score [75]. Another study found that the serial modified-LUS score in patients with severe COVID-19 could predict the need for prolonged mechanical ventilation lasting more than 21 days [76].

Other studies have compared LUS scores with various clinical parameters. One study introduced the Point-of-care ultrasound Lung Injury Score (PLIS) to predict the clinical course of COVID-19 patients [77]. The authors found that PLIS scores were directly associated with Sequential Organ Failure Assessment (SOFA) scores, and multivariate analysis revealed that each one-point increase in PLIS was linked to a higher risk of ICU admission and in-hospital mortality. Another ICU-based study demonstrated that LUS scores were significantly and inversely correlated with P/F ratios in mechanically ventilated COVID-19 patients [78]. In a large cohort of 229 patients with community-acquired pneumonia, C-reactive protein levels and LUS findings both declined between days 13 and 16 [79]. Among newborns with COVID-19, a strong positive correlation was observed between LUS scores and IL-6 levels [80]. These consistent correlations between LUS scores and inflammatory markers support the use of LUS as a tool for monitoring inflammatory lung diseases.

Finally, studies have revealed a high specificity of LUS in reflecting the evolution of pneumonia, even in patients with cardiac dysfunction [31] and acute myocardial infarction [32]. Therefore, when referenced against baseline findings, changes in LUS can offer valuable information throughout the treatment course in the HaH setting. As most current evidence is derived from studies on COVID-19 pneumonia, further validation is needed in patients managed within the HaH setting.

## 5. Conclusions

Clinicians aiming to diagnose and manage pneumonia within a HaH model must also develop the ability to recognize potential confounders from comorbidities and to differentiate sonographic patterns of pneumonia from other mimickers. In the second part of our review, we summarize evidence supporting the implementation of repetitive use of PoCUS throughout the course of pneumonia. Although much evidence arises from COVID-19, several studies on community-acquired pneumonia (CAP) have shown similar findings. Understanding the strengths and limitations of LUS in pneumonia care provides valuable insights for improving diagnostic accuracy and delivering person-centered care in HaH programs. In the subsequent part of our review, we will address the prognostic value of LUS, comparisons with CXR, and concerns regarding overdiagnosis in the HaH setting.

## Figures and Tables

**Figure 1 diagnostics-15-01200-f001:**
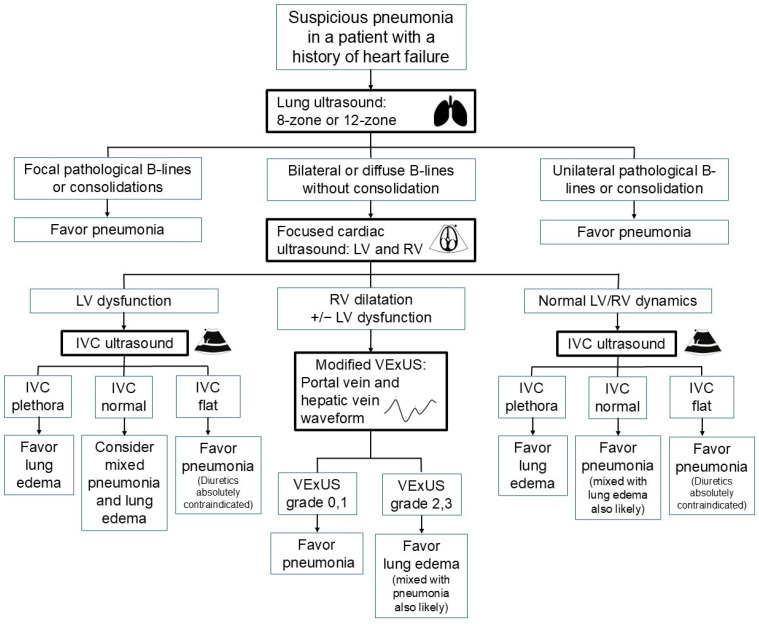
Flow diagram for evaluating pneumonia in patients with a history of heart failure (IVC, inferior vena cava; LV, left ventricle; RV, right ventricle; VExUS, venous excess ultrasound).

**Figure 2 diagnostics-15-01200-f002:**
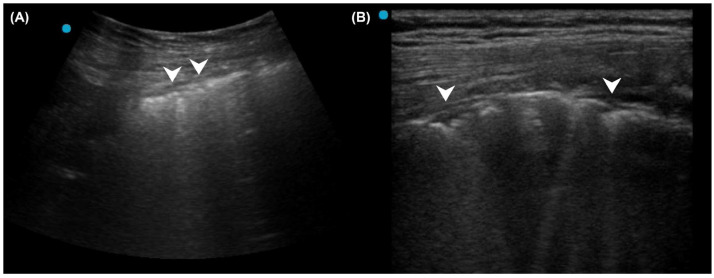
Pleural thickening and irregularity (arrowheads) in a patient with idiopathic pulmonary fibrosis: (**A**) Curvilinear probe; (**B**) High-frequency linear probe can help identify the uneven thickness of the pleura.

**Figure 3 diagnostics-15-01200-f003:**
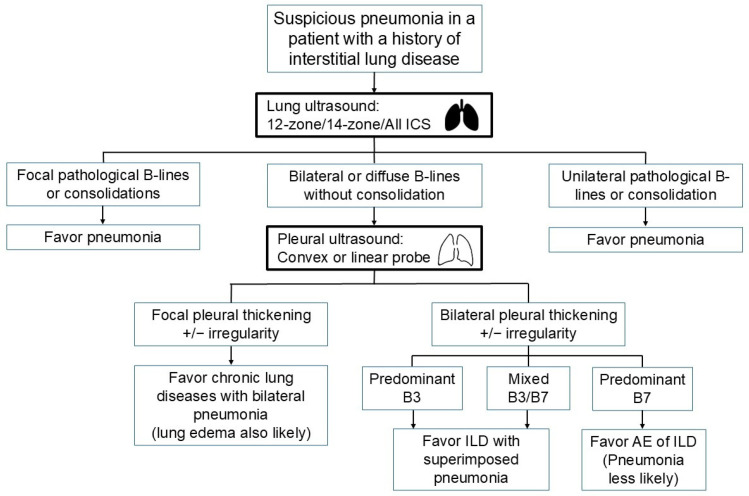
Flow diagram for evaluating pneumonia in patients with a history of interstitial lung disease (AE, acute exacerbation; ICS, intercostal space; ILD, interstitial lung disease).

**Figure 4 diagnostics-15-01200-f004:**
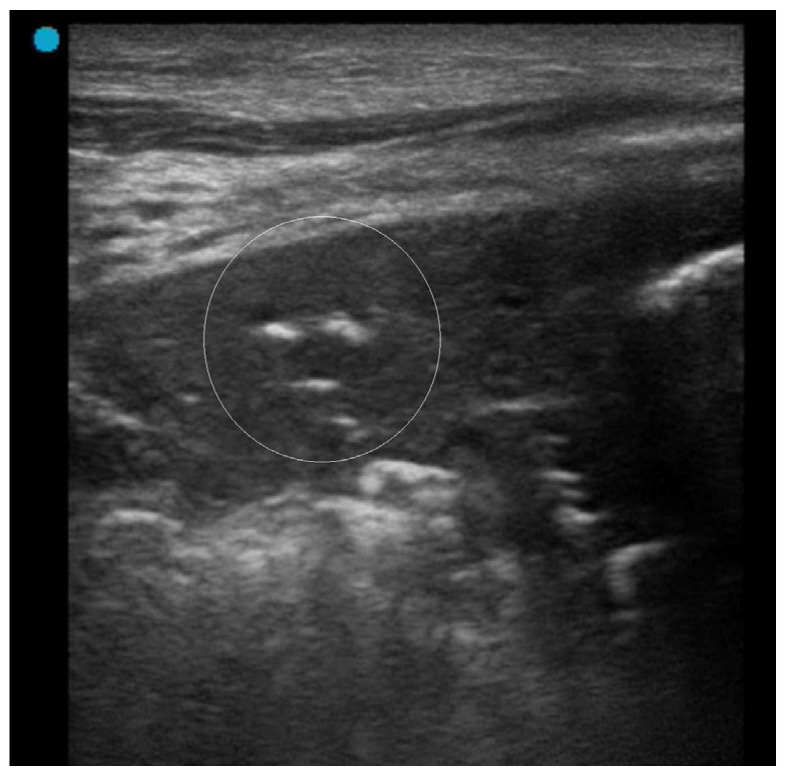
Static air-bronchogram (white circle) in a patient with right upper lobe atelectasis related to prior tuberculosis infection.

**Figure 5 diagnostics-15-01200-f005:**
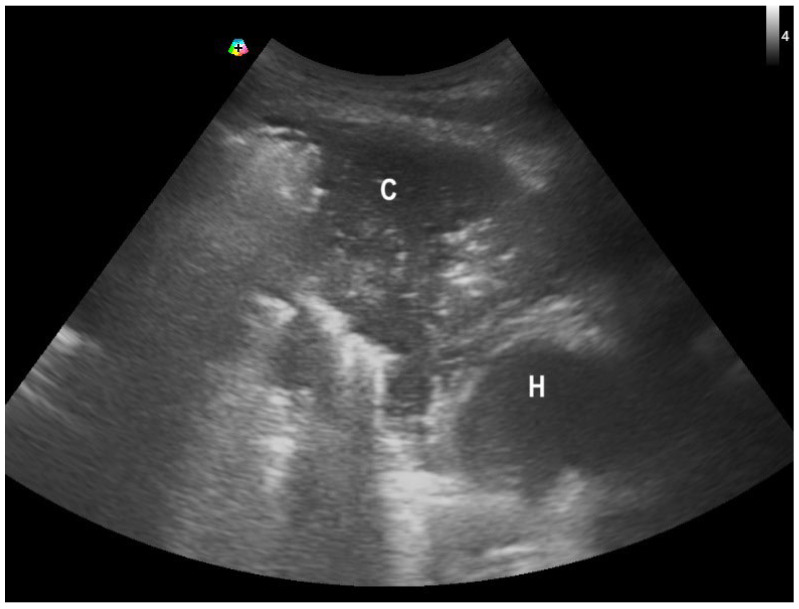
A patient with right middle lobe collapse with consolidation (C) along with a right deviation of the heart (H).

**Figure 6 diagnostics-15-01200-f006:**
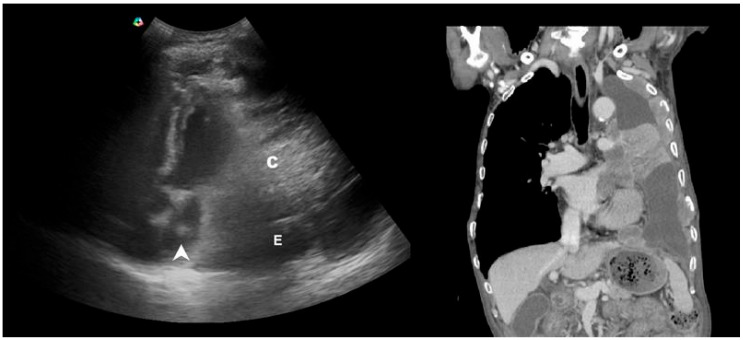
A patient with left lung cancer presenting with a large consolidative lesion (C) and massive effusion (E), with left atrial invasion (arrowhead) (LV, left ventricle).

**Table 1 diagnostics-15-01200-t001:** Ten essential questions to address before using point-of-care ultrasonography to diagnose and manage pneumonia in the hospital-at-home model.

1. What ultrasound techniques are essential for diagnosing pneumonia?
2. What are the ultrasound patterns associated with pneumonia?
3. Do different settings or etiologies of pneumonia influence the diagnostic accuracy of ultrasonography?
4. Do pulmonary comorbidities affect the accuracy of ultrasound diagnosis for pneumonia?
5. Do other differential diagnoses mimic the ultrasound patterns of pneumonia?
6. Do ultrasound findings correlate with pneumonia severity?
7. Do initial ultrasound findings associated with pneumonia hold prognostic value?
8. Do the ultrasound patterns improve in accordance with pneumonia recovery?
9. Is ultrasound superior to chest x-ray for diagnosing pneumonia?
10. Does ultrasonography lead to overdiagnosis of pneumonia?

## Data Availability

No new data were created or analyzed in this study.

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
