# Peer review of "Ten Questions on Using Lung Ultrasonography to Diagnose and Manage Pneumonia in Hospital-at-Home Model: Part II—Confounders and Mimickers"

_diagnostics, 2025, doi:10.3390/diagnostics15101200_

Round 1

Reviewer 1 Report

Comments and Suggestions for Authors

Dear Authors,

Many thanks for providing me the opportunity to review your manuscript:

"diagnostics-3601584_Ten Questions on Using Lung Ultrasonography to Diagnose and Manage Pneumonia in Hospital-at-Home Model: Part II. Confounders and Mimickers".

It is a well-organized and nicely presented manuscript that is trying to approach key clinical questions in the use of point-of-care thoracic ultrasound for pneumonia.

The manuscript comes in continuity with the previous review article and focuses on the proposed research questions that are relevant to the differential diagnosis and the severity of the disease.

The literature review is thorough and pertinent to the context and provides informative insights.

My only recommendation would be to consider making a few comments on how the presence of pleural reaction and/or pleural effusion could be an ancillary finding in the key consolidation-like features.

It is quite clear that your effort is trying to answer how the consolidative features, characteristic of pneumonia-type patterns, can vary within the range of differential diagnosis.

It would be a helpful addition to consider commenting on how the presence of these findings can be assisted by pleural effusions when they occur.

For example, in pneumonia with and without heart failure component, a pleural effusion can be present. If so,  would the pleural effusion +/- the pleura itself would help or confuse the diagnosis? In a pneumonia with pleural effusion present, the low or high echogenicity, the swirling sign, the fibrinous bands/septations/pockets can help, at some extent, answer the question if the consolidated or atelectatic parenchyma has developed due to inflammatory reaction/infection or because of heart failure. Equally,  when there are findings of B-lines,  accompanied by pleural effusions unilaterally or bilaterally, the inflammatory or fluid overload scenario could be somehow clarified, within the context of known limitations.

Again, I do understand that your focus is the parenchymal change per se, however since you are suggesting other ancillary findings [jugular distension, liver congestion, IVC features etc], the pleural effusion features could be of similar value.

Overall, I think your manuscript is valuable and well-presented with satisfactory supportive evidence.

Author Response

Dear Reviewer,

   We appreciate your kind recommendation which is valuable to improve our work. We are pleased to respond with some revisions.

  1. “My only recommendation would be to consider making a few comments on how the presence of pleural reaction and/or pleural effusion could be an ancillary finding in the key consolidation-like features.

It is quite clear that your effort is trying to answer how the consolidative features, characteristic of pneumonia-type patterns, can vary within the range of differential diagnosis.

It would be a helpful addition to consider commenting on how the presence of these findings can be assisted by pleural effusions when they occur.

For example, in pneumonia with and without heart failure component, a pleural effusion can be present. If so, would the pleural effusion +/- the pleura itself would help or confuse the diagnosis? In a pneumonia with pleural effusion present, the low or high echogenicity, the swirling sign, the fibrinous bands/septations/pockets can help, at some extent, answer the question if the consolidated or atelectatic parenchyma has developed due to inflammatory reaction/infection or because of heart failure. Equally, when there are findings of B-lines, accompanied by pleural effusions unilaterally or bilaterally, the inflammatory or fluid overload scenario could be somehow clarified, within the context of known limitations.

Again, I do understand that your focus is the parenchymal change per se, however since you are suggesting other ancillary findings [jugular distension, liver congestion, IVC features etc], the pleural effusion features could be of similar value.:

Response and revisions:

Thank you for pointing out that the presence of pleural effusions may also confound the recognition of consolidation (air bronchogram), B-lines, and pleural pathology. We think it is very important and add a subsection in our revised manuscript (Line 200-209).

2.3 Pneumonia diagnosis in the presence of pleural effusion

“It is important to note that both A-line and B-line artifacts rely on a pleural surface that is parallel and close to the chest wall and ultrasound probe. In the presence of pleural effusion, the visceral pleura is displaced away from the probe, becoming non-parallel to the chest wall and non-perpendicular to the ultrasound beam, which prevents the formation of A-lines. Furthermore, subpleural interlobular septa are no longer aligned with the ultrasound beam, hindering the generation of B-lines. Several other horizontal or vertical artifacts may appear but are not associated with pleural or alveolar pathol-ogy. Therefore, in cases of pleural effusion, only subpleural consolidations and air bronchograms remain reliable sonographic signs for diagnosing pneumonia.”

Reviewer 2 Report

Comments and Suggestions for Authors

The manuscript is well structured, timely, and clinically relevant. It extends the authors’ previous review by addressing Questions 4–6 (pulmonary comorbidities, mimickers, and severity/prognosis) and cites an impressive range of recent literature. The flow is generally clear, and the clinical algorithms (Figures 1 and 3) will be valuable to hospital-at-home (HaH) providers. That said, several clarifications and minor corrections are warranted before acceptance.

1. Introduction. Please add one sentence explicitly stating that Part I covered Questions 1–3 and that the present paper addresses Questions 4–6.

2. Question 6. The evidence cited here relies largely on ICU/COVID-19 cohorts. A brief statement noting that HaH-specific prospective data remain limited—and that the current conclusions are extrapolated from higher-acuity settings—would help readers interpret the findings appropriately.

3. Figure 2. Add arrows or dashed boxes to highlight the pleural thickening/irregularity. In the caption, explain what the arrows indicate and clarify how image A (curvilinear) and image B (high-frequency linear) differ in diagnostic clarity. Also correct the spelling to “curvilinear probe.”

4. Figure 4. Insert arrows or a circle around the hyperechoic static air-bronchogram, and note in the caption that its lack of respiratory motion distinguishes atelectasis from pneumonia.

These small additions will make the manuscript clearer for first-time HaH users of lung ultrasound while preserving its current strengths.

Author Response

Dear Reviewer,

We appreciate your sincere recommendations that we believe are valuable to improve our work. We are pleased to respond to each point with some revisions.

  1. Introduction. Please add one sentence explicitly stating that Part I covered Questions 1–3 and that the present paper addresses Questions 4–6.

Response and revisions:

Thank you for reminding us to include a summary of Part I to help readers better understand and follow the continuation. In the introduction section of the revised manuscript, we add (Line 48-52):

“In Part I of our review, we summarised current PoCUS practices for diagnosing pneumonia in ward and emergency department settings (Table 1). We reviewed key techniques and primary sonographic patterns that may be adapted for home care settings, and recommended a bilateral anterior, lateral, and posterior zone-based examination protocol.”

  1. Question 6. The evidence cited here relies largely on ICU/COVID-19 cohorts. A brief statement noting that HaH-specific prospective data remain limited—and that the current conclusions are extrapolated from higher-acuity settings—would help readers interpret the findings appropriately.

Response and revisions:

          Thank your for your comment. It is absolutely necessary to point out that most evidence are gathered in the COVID-19 era. We added this sentence in the revised manuscript (Line 347-349):

“As most current evidence is derived from studies on COVID-19 pneumonia, further validation is needed in patients managed within the HaH setting.”

  1. Figure 2. Add arrows or dashed boxes to highlight the pleural thickening/irregularity. In the caption, explain what the arrows indicate and clarify how image A (curvilinear) and image B (high-frequency linear) differ in diagnostic clarity. Also correct the spelling to “curvilinear probe.”

Response and revisions:

          Thanks for your comment to improve the readability. We add arrowheads to the Figure 2, and also revised the figure legend as: “Pleural thickening and irregularity (arrowheads) in a patient with idiopathic pulmo-nary fibrosis: (A) Curvilinear probe; (B) High-frequency linear probe can help identify the uneven thickness of the pleura.”

  1. Figure 4. Insert arrows or a circle around the hyperechoic static air-bronchogram, and note in the caption that its lack of respiratory motion distinguishes atelectasis from pneumonia.

Response and revisions:

          Thank you for the comment that improve the readability of or figure. We are pleased to add a white circle to Figure 4, and revise the caption as follows.

“Static air-bronchogram (white circle) in a patient with right upper lobe atelectasis related to a prior tuberculosis infection.”